# The Effect of Colostrum Supplementation during the First 5 Days of Life on Calf Health, Enteric Pathogen Shedding, and Immunological Response

**DOI:** 10.3390/ani14081251

**Published:** 2024-04-22

**Authors:** Anna Catharina Berge, Iris Kolkman, Pleun Penterman, Geert Vertenten

**Affiliations:** 1Veterinary Epidemiology Unit, Department of Internal Medicine, Reproduction and Population Medicine, Faculty of Veterinary Medicine, Ghent University, Salisburylaan 133, 9820 Merelbeke, Belgium; 2A7 Noord Dierenartsen, 9207 JA Drachten, The Netherlands; 3MSD Animal Health, Wim de Körverstraat 35, 5831 AN Boxmeer, The Netherlands

**Keywords:** colostrum supplementation, post-closure colostrum, calf enteric pathogens, colostral immunity

## Abstract

**Simple Summary:**

This farm trial investigated the effect of post-closure colostrum supplementation on the preweaning health, performance, immunity, and enteric pathogen shedding of preweaned dairy calves. Colostrum from dams vaccinated against rotavirus, coronavirus, and *Escherichia coli* F5 and F41 was collected and frozen. The colostrum supplement was fed once daily to 39 calves (Colost-suppl.) and a supplement in similar nutritional composition as the colostrum but devoid of immunoglobulins was fed similarly to 36 Control calves for 4 days post-closure of the gut (at 24 h after birth). Fecal samples and serum samples were collected on day 2, day 7, day 14, and day 21 for the detection of enteric pathogens and serum pathogen-specific and total immunoglobulin G levels. Calves were inspected daily, and their clinical health was scored. There were no significant differences in pathogen-specific and total serum immunoglobulin levels, fecal scores, and weight gains between the treatment groups. There were significantly less respiratory clinical signs in Colost-suppl. compared to Control calves. The fecal samples collected were all negative for coronavirus and *E. coli* F5. At 14 days of age, there were significantly less *Cryptosporidia*-positive samples and a non-significant trend for fewer rotavirus-positive samples in Colost-suppl. calves compared to Control calves.

**Abstract:**

The objective of this dairy farm study was to investigate the preweaning health, performance, immunity, and enteric pathogen shedding in calves supplemented with colostrum during five days after birth compared to calves not supplemented with colostrum. The colostrum supplementation was the previously frozen colostrum added to the milk replacer from day 2–5 given to 39 calves, and 36 Control calves received a milk-derived supplement. There was no significant difference in preweaning weight gain between the treatment groups. Serum samples collected on days 2, 7, 14, and 21 indicated that total and antigen-specific IgG levels against rotavirus, coronavirus, and *E. coli* F5 were not significantly different between the treatment groups. Fecal samples taken on days 7, 14, and 21 were all negative for coronavirus and *E. coli* F5, whereas there were low levels of *Cryptosporidia* and a trend for low levels of rotavirus on day 14 in colostrum-supplemented compared to Control calves. Respiratory clinical signs, depressed attitude and body temperature tended to be reduced in colostrum-supplemented compared to Control calves. This study shows that, even in calves with good colostrum status and high plane of nutrition, there can be benefits of post-closure colostrum supplementation including reduced *Cryptosporidia* and rotavirus shedding and reduced respiratory disease.

## 1. Introduction

The prophylactic and therapeutic effects of colostrum to reduce diarrhea in preweaned calves should be implemented by the dairy industry to reduce antimicrobial use and antibiotic resistance in young stock [1].

Colostrum is a powerful nutritious meal containing macronutrients, micronutrients, immunoglobulins, bioactive compounds, and whole immune cells [1,2]. Colostrum with a high level of immunoglobulins needs to be administered as soon as possible after birth to provide optimal systemic immunity. The strong emphasis on the first colostrum feeding is critical to reduce morbidity and mortality in calves. There is a continual transfer of serum immunoglobulins, such as rotavirus antibodies, obtained through the first colostrum feed to the intestinal lumen, which provides protection against enteritis [3,4]. Despite this well-known fact, many dairy calves still have sub-optimal immunity due to not have been fed sufficient timely colostrum after birth [5]. The systemic passive transfer of immunity (transfer of immunoglobulins from colostrum into the blood of the calf) gradually decreases over the first 24 h to close to 10% of initial absorptive capacity [6]. Many dairy calves are directly fed whole milk, waste milk, or milk replacer after the first colostrum feeding, and calf management advice has placed less emphasis on continued colostrum administration or transition milk after the first critical 24 h.

Colostrum contains many beneficial bioactive compounds, growth factors, and hormones that are involved in the maturation of the intestinal tract and repair functions [7,8,9,10]. Early research indicated that colostrum or transition milk feeding provides local gut immunity against enteric viral and bacterial pathogens, reduces the effect of bacterial enterotoxins, and may enhance intestinal villus development [11,12,13,14]. Over 25 years ago, it was already demonstrated that neonatal calves fed colostrum exhibited six times greater villus circumferences, areas, and heights in total small intestine and enhanced absorptive capacity compared to calves fed only milk replacer [14,15]. Feeding high amounts of first colostrum for more feedings has been shown to promote epithelial cell growth and enhance the survival of mature mucosal epithelial cells [7]. In addition, colostrum feeding has a significant impact on the glucose status of the neonatal calf and colostral bioactive substances, such as hormones and growth factors, may contribute to neonatal glucose homeostasis [16].

There are now several studies evaluating the effect of post-closure colostrum feeding on calf health. A multi-farm study of post-closure colostrum supplementation of milk replacer to pre-weaned calves on California calf ranches during the first 14 days of life reduced diarrheal disease days by 40% [17]. Several other studies have indicated that 14-day colostrum supplementation reduced diarrhea and improved weight gain [18,19]. A recent study indicated that colostrum supplementation for four days may be an effective therapy for diarrhea in preweaning calves and thereby improving preweaning weight gain [20]. Thus, it is likely that these long-term supplementations also had some therapeutic effects. Hyperimmune colostrum is available on all farms vaccinating the dams in the dry periods for calf pathogenic diseases, and it is important to find feeding systems that optimize the use of this valuable feed with immune-enhancing properties for calves. Hyperimmune colostrum derived from cows exposed to calf pathogens can also be used as therapy for calf diarrhea [1]. Field studies need to determine optimal quantities and practical approaches for colostrum supplementation. Furthermore, the gut protective effects of post-closure colostrum supplementation in terms of pathogen-shedding need to be explained.

The aim of this study is to evaluate the benefits of colostrum supplementation of feedings to calves post-closure of the gut to 5 days of age on health, performance, enteric pathogen shedding, and immunity, to be able to determine a cost-effective manageable system to prevent the most prevalent and costly health challenges in heifer rearing. 

## 2. Material and Methods

This study was carried out in preweaned calves on one commercial Holstein-Friesian dairy farm with a lactating herd size of 220 cows in Eastermar Friesland in the Netherlands. The total average milk yield per cow (305 days) was 10,500 kg. The average age of the first calving was 24.5 months, and the replacement rate was 23%, with all internal recruitment. The dams were all vaccinated against calf enteric pathogens rotavirus, coronavirus, and *Escherichia coli* F5 and F41 within 6 weeks prior to calving (Bovilis^®^ Rotavec^®^ Corona, MSD Animal Health). The farm had a history of cryptosporidiosis in the calves. 

The study was carried out by farm personnel and the herd veterinary practice (denoted study personnel) as a part of the regular veterinary herd health contract. The calves were removed within 2 h after birth from the calving pen and fed two colostrum feeds using nipple buckets. The calves were raised outdoors in individual Plexiglas hutches for the first month of life (internal size:2.0 by 1.2b y 1.3 m; external size: 2.0 by 1.2 m) and were thereafter transferred to pen groups of 3 calves per pen (3 by 5 m). Weaning was performed over a period of one week at approximately 3 months of age. The calves were fed milk replacer containing 23% protein and 17% fat, with 50% skim milk and 100% milk proteins fed at 150 g milk replacer per liter from the second day of life until weaning. The calves were fed twice daily, at 7 and 18 o’clock, 3.5 L per feeding in the first week, and 4.5 L per feeding from second week of life to 10th week of life, and thereafter, a gradual reduction in quantities fed until week 14, when 2 L once per day were fed for one week prior weaning at 15 weeks of age. The calves were further fed ad libitum starter muesli grain from 10 days of age until 4 weeks of age and were thereafter fed an all-pelleted grain supplemented with hay until weaning. Water was available ad libitum.

The colostrum supplement was the excess first milking colostrum collected prior to and during the study, bagged in plastic zip-lock bags in quantities of 1 L and 0.5 L, and frozen at −18 °C. The control supplement was a skim milk-derived supplement with similar fat and protein composition and color as the colostrum supplement (devoid of immunoglobulins), which was similarly bagged and frozen as the colostrum supplement. The control supplement and colostrum supplement were coded 1 or 2, to allow for the blinding of study personnel performing feeding and calf health evaluations. A small sample (50 mL) from every colostrum collection, with information of cow-id and date on sample container were frozen.

Calves that were physically healthy at birth were eligible for enrolment into the study from all parity cows. Seventy-five calves were enrolled from 20 November 2019 to 23 March 2023, and 39 calves were randomly allocated to Colost-suppl. and 36 to Control treatment. Weights were recorded on day 0, at 4 weeks of age (±2 days), and at weaning at around 3 months of age (range: 78–112 days). All calves were fed two meals of colostrum in accordance with farm protocols, with a mean for the first colostrum feeding of 3.8 Liters and in the first 24 h on average of 8.9 Liters. A sample of first colostrum fed to study calves was collected, marked with cow-id and calf-id, and frozen. A birthday document included dam-id, parity, calving time, calving ease, calf-id, time of milking of colostrum, quantity of colostrum, quality of colostrum (Brix %), time of colostrum feed, and quantity of colostrum fed to the calf. The calves received 1 L of colostrum or placebo supplement in the morning milk replacer feeding 24 h after birth and half a liter of colostrum or placebo in the morning milk replacer feeding for the next 3 days. The preweaning milk, milk replacer, and grain and water feeding were similar for both groups. Milk feed intakes were monitored and recorded during the first week. Calf concentrate feed and water intake was ad libitum and not recorded.

Calves were independently evaluated daily for the first 21 days by study personnel for morbidity (fecal scores, respiratory scores, attitude scores, and other observations) using a daily health scoring sheet as previously described [21]. Calves were evaluated for body temperature daily in the first 3 weeks and thereafter in those showing signs of disease when older than 3 weeks. Treatments were performed and recorded by on-farm personnel based on criteria established and monitored by the veterinary practice.

Total serum IgG concentrations were determined on the second day when the calf was older than 24 h to evaluate the transfer of passive immunity using an ELISA quantitative methodology (BIO-X Diagnostics, Rochefort, Belgium, BIO K 165). Serum samples taken at 1, 7, 14, and 21 days of age were used to determine antibody levels to bovine coronavirus, rotavirus, and *E. coli* F5, with competitive enzyme-linked immunosorbent assays (ELISA) (BIO-X Diagnostics, Belgium, BIO-K 126, BIO-K 295, BIO-K 392).

Fecal samples were taken at 7, 14, and 21 days of age and on the first day of diarrheal disease diagnosis (fecal score ≥ 2). The presence of fecal pathogens was determined using rapid ELISA kits (rotavirus, coronavirus, *Cryptosporidium parvum*, *Clostridium perfringens*, *Escherichia coli* F5) (BIO-X Diagnostics, Belgium BIO-K 306).

Daily records were maintained of all management and clinical health observations and tests performed in the calves. These daily records were photographed and transmitted to the study director for further data entry and analysis. 

### Statistical Analysis

The null hypothesis in this study was that there is no significant difference in outcome parameters between Colost-suppl. calves and Control calves. The alternative hypothesis was that the Colost-suppl. calves have significantly different outcomes from the Control calves.

Initial sample size calculations indicated that with 36 calves per group, we could detect a significance (α = 0.05) from 12% to 8% in diarrhea-days in the first 3 weeks between the treatment groups with a power of 80%. The experimental unit was the calf.

The photographed data records and all reports from laboratory analysis were obtained by the study epidemiologist and statistician (first author). The data were entered into Excel and double checked for accuracy prior to analysis. The statistical analysis was performed in SAS version 9.4. All continuous variables were evaluated for normality, and summary statistics were created including the number of calves (N), the mean, the standard deviation, the minimum and the maximum values, and the range. Differences between the outcomes of the Control compared to Colost-suppl. calves were statistically compared in univariate linear models. The statistical level of significance was set at *p* ≤ 0.05 and a non-significant trend was declared at 0.05 < *p* ≤ 0.10.

The serum IgG levels on day two were classified as poor, fair, good, and excellent passive transfer of immunity according to Lombard et al. [5]. Daily clinical health evaluations were evaluated using stratified analysis, and differences between colostrum-supplemented and Control calves were assessed with Chi-square tests and the non-parametric Jonckheere–Terpstra test of trend. When differences were observed in the univariate analysis between treatments with a *p* ≤ 0.10, further analysis was performed using generalized estimating equation (GEE) models with repeated measures on calf. For binomial outcomes, binary logistic regressions were used, and for ordinal outcomes, a cumulative logistic regression was used. For body temperature, a mixed model with a random effect of calf was used. These models controlled for other covariates such as serum IgG concentrations on day 2, parity, season, and birth weight, and covariates were retained when *p* ≤ 0.10. The number of days to diarrhea was evaluated with survival analysis using Kaplan–Meier plots and log-rank statistical tests.

Microbial fecal rapid tests performed on days 7, 14, and 21 were evaluated with stratified analysis, and Chi-squared tests were used to determine differences between Colost-suppl. versus Control groups for the binary outcomes (rotavirus, coronavirus, *E. coli* F5, and *C. parvum*) and the non-parametric Jonckheere–Terpstra test of trend was used for *C. perfringens* fecal log counts.

Colostral IgG and serum IgG levels in mg/mL were compared at the sampling times between treatment groups in general linear models. Percentage inhibition in ELISAs for antibodies to rotavirus, coronavirus, and *E. coli* F5 at the sampling times were compared in general linear models.

## 3. Results

### 3.1. General Study Results

The overall health of the calves during the study was in general good, and no disease outbreaks were noted.

There was no significant difference in colostrum intake during the first 24 h, serum IgG levels, milk intake in the first week, or weight gain in the Colost-suppl. calves and Control calves (Table 1). The passive transfer of immunity in all calves assessed through serum IgG levels on day 2 was on average very good and not significantly different between Colost-suppl. calves (21.3 mg/mL) and Control calves (21.7 mg/mL) (*p* = 0.95). The milk intake in the first week was not significantly different between Colost-suppl. calves (44.0 L) and Control calves (42.8 L) (*p* = 0.23), and the appetite was in general very good with an average appetite score of 2.0 in Colost-suppl. and 1.9 in Control calves (*p* = 0.12). The birth weight was not significantly different between Colost-suppl. (42.2 kg) and Control calves (41.9 kg) (*p* = 0.69). The average daily weight gain during the first 4 weeks was not significantly different between the Colost-suppl. calves (0.82 kg/day) and the Control calves (0.86 kg/d) (*p* = 0.31). The average weight daily gain from 4 weeks to weaning was 1.08 kg/day for the Colost-suppl. calves and not significantly different from 1.10 kg/d in Control calves (*p* = 0.59). The average daily weight gain from birth to weaning was 0.99 kg/day for the Colost-suppl. calves and not significantly different from 1.02 kg/d in Control calves (*p* = 0.39).

### 3.2. Non-Parametric Analysis of Clinical Health Observations

Clinical health observations were evaluated using stratified analysis and non-parametric statistical testing (Table 2). There were significantly higher respiratory scores in Control compared to Colost-suppl. calves in the first week and the whole preweaning period (*p* < 0.01). There was a non-significant trend to more depressed attitude observations in Control compared to Colost-suppl. calves in the preweaning period (*p* = 0.08). Navel scores tended to be non-significantly higher in Control compared to Colost-suppl. calves in the preweaning period (*p* = 0.07). There were a few other miscellaneous observations that scored higher in Control calves (*p* = 0.04); however, they had spurious statistical significance and were not analyzed further. Body temperatures were significantly lower in Colost-suppl. compared to Control calves (*p* < 0.01) (Table 2).

The number of days to first diarrhea (fecal score ≥ 2) were evaluated with survival analysis, and Kaplan–Meier plots indicated that there was no significant difference in days to diarrhea between Control and Colost-suppl. Calves (*p* = 0.97) (Figure 1).

The GEE logistic models, incorporating repeated measures on calf, were further evaluated for respiratory scores, navel disorder, and attitude and indicated that no other predictive variables were significant (Table 3). The models estimated a non-significant 72–73% reduced risk of respiratory clinical signs in first week of life and in the preweaning period in Colost-suppl. compared to Control calves (*p* = 0.14 and *p* = 0.11, respectively). The GEE model of attitude indicated a non-significant 55% reduced risk of depressed attitude in Colost-suppl. compared to Control calves (*p* = 0.11) (Table 3). There was no significant difference in navel pain/swelling between Colost-suppl. and Control calves (*p* = 0.53). The mixed model (including a random effect of calf and the IgG status of the calf as a covariate) indicated that there was a non-significant tendency for lower body temperatures in Colost-suppl. calves compared to Control calves (*p* = 0.10) (Table 4).

### 3.3. Microbial Fecal Tests

Microbial fecal rapid tests indicated the reductions in fecal positive samples for *C. parvum* and rotavirus at 14 days of age in Colost-suppl. compared to Control calves (Table 5 and Figure 2). There were no positive findings of *E. coli* F5 and only one finding of coronavirus in one calf on day 14. There were significantly less *C. parvum*-positive fecal samples in Colost-suppl. compared to Control calves on day 14 (*p* = 0.04). There was a non-significant trend for lower levels of rotavirus in Colost-suppl. compared to Control calves on day 14 (*p* = 0.09). The *C. perfringens* log counts were not significantly different between treatment groups (*p* = 0.23).

### 3.4. Serum Antibody Levels

Serum IgG was 21.3 mg/mL in Control calves on day 2, decreasing to 14.7 mg/mL on day 21, and was 21.4 mg/mL in Colost-suppl. calves on day 2, decreasing to 15.5 mg/mL on day 21 (Table 6). There were no significant differences in serum IgG at any sampling time. Percentage inhibition in indirect ELISA for coronavirus and *E. coli* F5 showed constant elevated values, whereas there was a reducing trend for rotavirus % inhibition values. There was no significant difference between the groups at any day for the three pathogens tested (Table 6, Figure 3).

## 4. Discussion

This study evaluating the effect of 5 days of colostrum supplementation of calves on the health and immunological status of the calves was performed in a dairy herd where the overall colostrum management of the calves was good, the passive transfer of immunity was high, and the overall health of the calves was good, and the herd was administered rotavirus, coronavirus, and *E. coli* F5 and F41 vaccination of the dry cows. High systemic colostral immunity has been shown to improve calf health in numerous studies [6]. The high passive transfer of immunity in our study provided a good immunoglobulin gut protection against enteric pathogens in both control and post-closure colostrum-supplemented calves, thus reducing the possible detectable clinical differences in this relatively small study. Over 30 years ago, it was shown that serum-derived antibodies persist in the gut of the calf between milk feeds, whereas milk-derived antibodies do not persist as long between milk feeds, when the calves are fed twice daily, but they both contribute to intestinal immunity [3]. In our study, the calves were fed milk replacer supplemented with colostrum once daily; thus, this most likely produced a more time-limited gut-local colostral immunoglobulin protection coverage compared to calves that obtained colostrum supplements in the feed twice daily. Our study cannot determine the components of colostrum that enhance the post-closure gut health, and it is highly likely that various colostrum components such as cytokines, growth promoters, and oligosaccharides are involved [2].

In this study, due to the limited sample size of calves on a commercial dairy enrolled over a long period, we have chosen to regard differences between Colost-suppl. calves and Control calves with a statistical *p*-value of ≤0.10 as a non-significant tendency of clinical importance. The long duration of the study may introduce variability in outcomes due to seasons and possibly other management factors; however, both treatment groups were equally present in all seasons. The farm was requested to maintain the same management and feeding during the study period. The study personnel, i.e., the veterinary clinic located close to the farm, ensured consistency in observations and documentation over time. Due to financial constraints and duration of the study, we could not continue the trial simply to obtain more statistical significance.

### 4.1. Weight Gain

Overall high feeding levels with a high-quality milk replacer contributed to a high average daily gain of around 1 kg per day in the calves in the preweaning period, and no significant increase in weight gain was seen in the colostrum-supplemented calves, as have been described by other studies. One of the first studies on colostrum supplementation compared Swiss calves being fed colostrum/transition milk in the first 3 days compared to milk replacer and noted an improved preweaning weight gain in the colostrum and the transition milk-supplemented calves; however, in this study, the control calves did not receive any colostrum on the first day of life [22]. A study in Michigan, IN, USA, showed that three days of feeding transition milk or colostrum-supplemented milk to dairy calves increased preweaning weight gain by 3.5 kg compared to non-supplemented calves [23]. Iranian studies have also demonstrated that both colostrum and transition milk supplementation of pasteurized waste milk positively impacted health and growth of calves [19,24]. A recent Canadian study indicated that both 2 days and 14 days of post-closure colostrum supplementation (with a dried colostrum replacer) could improve the daily weight gain in the first three weeks of life [25]. The lack of effect of weight gain in our colostrum-supplemented was most likely due to the overall high level of gut health in the calves so that digestive functions were not impacted. 

### 4.2. Gut Health and Fecal Enteric Pathogens

We did not detect any significant differences in gut health scores in the Colost-suppl. calves compared to the Control calves during the preweaning period, and fecal scores indicated an overall good gut health. Nonetheless, we noted significantly higher prevalence of *C. parvum*-positive samples in Control calves at 14 days of age compared to Colost-suppl. calves. Furthermore, there was a non-significant trend for increased number of rotavirus-positive fecal samples in Control calves compared to Colost-suppl. calves at 14 days of age and no Colost-supp. calves were found positive for rotavirus. A rotavirus challenge study where calves were inoculated with rotavirus and fed a hyperimmunized colostrum supplement for 14 days, indicated that the calves were protected against rotavirus diarrhea for the first 21 days, reduced virus shedding, and improved their mucosal immunity [18]. This indicates that post-closure colostrum supplementation may be of value to reduce enteric diseases due to rotavirus and *Cryptosporidium parvum*. With the lack of effective treatments for cryptosporidiosis- and rotavirus-associated diarrhea, the ability of post-closure colostrum supplementation to alleviate this challenge for many dairy calves may be a very important preventive strategy. We were planning to analyze the fecal samples further for microbiome changes, but the limited impact on enteric health may not warrant this analysis.

### 4.3. Respiratory Signs

In our study, we noted a significant reduction in respiratory scores in the stratified analysis and a non-significant trend in the repeated-measure cumulative logistic models for reduced respiratory clinical signs (snotty noses and increased respiratory rates) in colostrum-supplemented calves. The reduction in respiratory scores was primarily due to a significant reduction in snotty noses and a reduction in the number of calves with an elevated respiratory rate. Like our study, an Irish study of transition milk feeding found no increase in serum IgG concentrations in calves but less respiratory disease signs due to the transition milk feeding [26]. A study of using colostrum supplementation as a nutraceutical intervention when decreased milk intake was recorded in calves indicated that the colostrum supplementation was associated with reduced lung lob consolidations and a reduced hazard of respiratory disease in the week following the intervention [27]. Numerous studies have noted an increased respiratory disease challenge in dairy calves that have experienced diarrhea [28,29]. Despite not noting differences in increased fecal scores in the Colost-suppl. calves compared to Control calves, we noted increased number of Control calves shedding rotavirus, *C. parvum,* and non-significantly increased log counts of *C. perfringens*, indicating that there may have been an elevated gut health challenge in the Control calves compared to the Colost-suppl. calves that could be associated with increased respiratory disease signs. 

### 4.4. Body Temperature and Attitude

An overall significantly increased higher body temperature was noted in the Control calves compared to the colostrum-supplemented calves. Although this effect was small overall, it was associated with days of fever or elevated temperatures at times when other clinical signs were noted. This trend may have been due to a slightly higher infectious pressure in the Control calves, possibly linked to the increased upper respiratory disease challenges, or a higher infection pressure due to rotavirus or *Cryptosporidia*. There was also a non-significant increased number of days that Control calves showed depressed attitude compared to the colostrum-supplemented calves, which may be due to mild sub-clinical disease. In a study of 14 days of supplementation of colostrum at 350 and 700 g per day, it was similarly noted that colostrum-supplemented calves had fewer days with fever and depressed attitude compared to non-supplemented Control calves [18].

### 4.5. Serum Antibodies

Serum IgG concentrations and serological antibodies to rotavirus, coronavirus, and *E. coli* F5 in the calves were not significantly different between colostrum-supplemented calves and Control calves. This indicated that post-closure colostrum supplementation had no significant influence on the systemic antibody levels in the Colost-suppl. compared to Control calves. This was similarly noted in the study by Conneely et al. in Ireland where they did not detect any increase in serum IgG concentrations in calves fed more transition milk [26]. Another study has shown that an additional five days of colostrum feeding positively influenced non-specific humoral immunity indicators and serum biochemical parameters in dairy calves [30]. Due to the limited disease challenges in this study, it was not deemed necessary to evaluate these study serums further for other immunity parameters.

## 5. Conclusions

This study has shown that, even in calves with a high passive transfer of immunity status, there are detectable benefits in the post-closure colostrum supplementation of calves for 4 days. The study indicated that, when previously frozen farm colostrum was added to the calf milk replacer once daily, it resulted in reduced *Cryptosporidium parvum* fecal shedding and a trend towards the lower shedding of rotavirus in calves 14 days of age in colostrum-supplemented calves compared to non-supplemented calves. Respiratory clinical signs, depressed attitudes, and body temperatures in calves under one month of age were reduced in colostrum-supplemented calves, and this may be associated with improved digestive system health and immunity.

## Figures and Tables

**Figure 1 animals-14-01251-f001:**
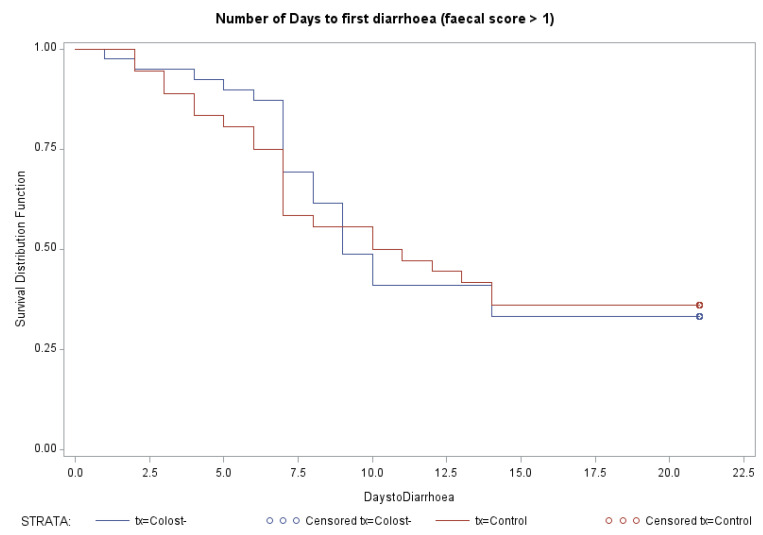
Kaplan–Meier plot for the number of days until first diarrhea in Colost-suppl. Calves (blue) and Control calves (red). The Control calves (TX = Control in red) and colostrum-supplemented calves (TX = Colost in blue).

**Figure 2 animals-14-01251-f002:**
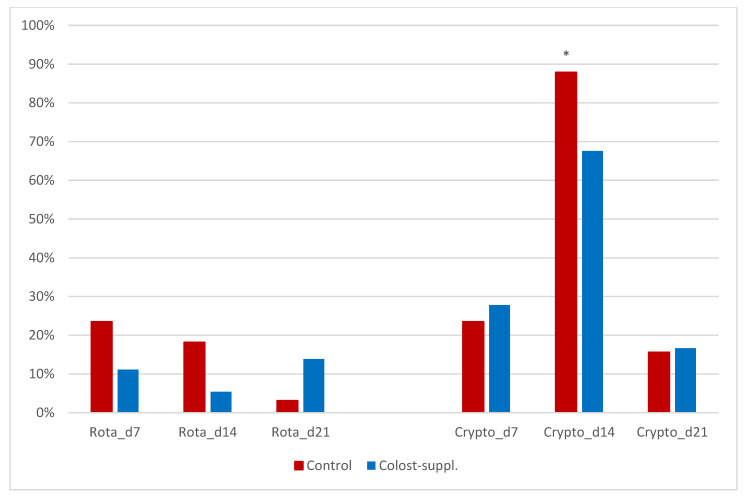
Histogram of % fecal samples that were found positive for rotavirus and *Cryptosporidium parvum* in Colost-suppl. and Control calves on days 7, 14, and 21. * = statistically significant difference in stratified analysis.

**Figure 3 animals-14-01251-f003:**
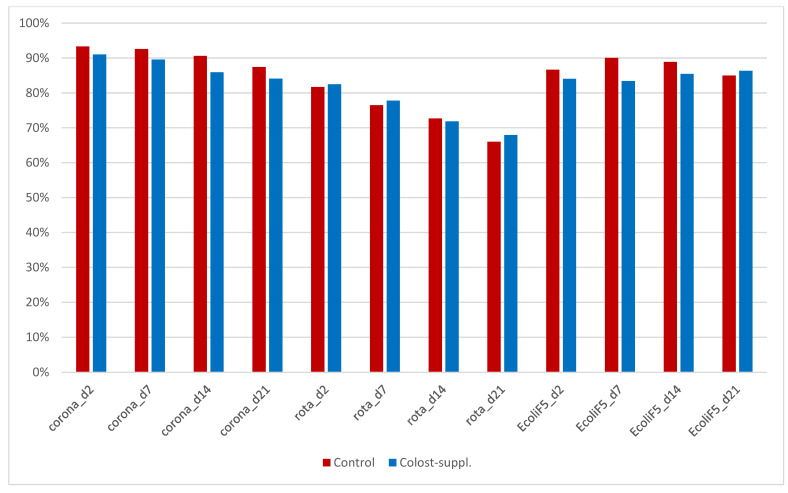
Histogram of % inhibition in indirect ELISA for coronavirus, rotavirus, and *E. coli* F5 in serum on days 2, 7, 14, and 21 for Colost-suppl. compared to Control calves.

**Table 1 animals-14-01251-t001:** Colostrum and milk intake and preweaning weight gain in. Control and Colost-suppl calves.

First Week	Control			Colost-Suppl.			Diff	*p*-Value *
	N	Mean	Std Dev	N	Mean	Std Dev		
Parity	36	3.53	1.9	39	3.41	1.5	−0.12	0.77
Birth weight (kg)	36	41.9	3.0	39	42.21	3.5	0.31	0.69
First colostrum feed (L)	36	3.8	1.0	39	3.7	1.0	−0.1	0.64
Colostrum (L) in first 24 h	35	9.0	1.4	38	8.9	1.7	−0.1	0.79
Colostrum %brix	34	23.8	2.7	39	24.1	2.5	0.3	0.65
Serum IgG at day 2 (mg/mL)	36	21.7	4.3	39	21.3	4.7	−0.37	0.95
Milk intake (L) in first week	36	42.8	5.4	39	44.03	3.1	1.23	0.23
Week 1’s appetite score	36	1.9	0.2	39	2.0	0.1	0.06	0.12
ADG birth—4 weeks	36	0.86	0.19	39	0.82	0.17	−0.04	0.31
ADG 4 weeks—weaning	34	1.10	0.23	37	1.08	0.18	−0.02	0.59
ADG birth—weaning	34	1.02	0.16	37	0.99	0.12	−0.03	0.39

* Differences between groups tested with general linear models.

**Table 2 animals-14-01251-t002:** Clinical health observations in the first week of life and in the whole preweaning period in Control calves and Colost-suppl. calves.

First Week	Control			Colost-Supp.			Diff	*p*-Value *
	N	Mean	Std Dev	N	Mean	Std Dev		
Temperature	247	38.9	0.4	267	38.8	0.4	−0.08	0.01
Fecal	248	0.5	0.8	268	0.5	0.6	−0.04	0.94
Hydration	252	0.0	0.1	271	0.0	0.1	−0.01	0.04
Diarrhea	248	0.1	0.3	268	0.1	0.2	−0.02	0.45
Respiratory	252	0.2	0.5	271	0.1	0.3	−0.1	<0.01
Attitude	253	0.0	0.1	271	0.0	0.1	−0.01	0.42
Eye	252	0.0	0.1	271	0.0	0.1	0.0	0.71
Ear	252	0.0	0.1	271	0.0	0.0	0.0	0.3
Navel	245	0.2	0.4	271	0.2	0.4	0.0	0.82
Joint	252	0.0	0.2	271	0.0	0.0	−0.02	0.14
Other	134	0.0	0.2	138	0.0	0.0	−0.03	0.04
Preweaning	N	Mean	Std Dev	N	Mean	Std Dev	Diff	*p*-value *
Temperature	339	38.9	0.4	378	38.8	0.4	−0.07	0.02
Fecal	357	0.7	0.9	397	0.8	0.9	0.06	0.32
Hydration	350	0.0	0.1	389	0.0	0.1	0.00	0.32
Diarrhea	357	0.1	0.4	397	0.2	0.4	0.01	0.75
Respiratory	350	0.1	0.5	386	0.1	0.3	−0.06	<0.01
Attitude	357	0.0	0.2	388	0.0	0.1	−0.02	0.07
Eye	351	0.0	0.1	387	0.0	0.1	0.00	0.57
Ear	350	0.0	0.1	386	0.0	0.0	0.00	0.29
Navel	343	0.2	0.4	386	0.1	0.4	−0.05	0.07
Joint	350	0.0	0.2	386	0.0	0.0	−0.01	0.14
Other	134	0.0	0.2	141	0.0	0.0	−0.03	0.04

* Differences between control and colostrum-supplemented groups tested with general linear models for temperature, and with stratified analysis and the non-parametric Jonkheere–Terpstra test of trend for the health scores.

**Table 3 animals-14-01251-t003:** Four separate GEE logistic regression models evaluating the odds of health disorders in Colost-suppl. compared to Control calves.

Model	Treatment	OR	Lower CI	Higher CI	*p*-Value
Resp * first week	Colost-suppl.	0.28	0.05	1.54	0.14
Resp * preweaning	Colost-suppl.	0.27	0.05	1.35	0.11
Attitude ** preweaning	Colost-suppl.	0.45	0.17	1.19	0.11
Navel *** preweaning	Colost-suppl.	1.43	0.47	4.37	0.53
All four models	Control	1.00	1.00	1.00	ref

* Resp = respiratory disease symptoms, ** Attitude = depressed attitude, and *** Tender navel.

**Table 4 animals-14-01251-t004:** A mixed model with a random effect of calf, evaluating the daily body temperatures in calves in the first 3 weeks in Colost-suppl. compared to Control calves.

Variable	Estimate	S.E.	*p*-Value	95% Conf. Interval	
				Lower CI	Higher CI
Colost-suppl.	−0.08	0.05	0.10	−0.17	0.01
Control	0.00	0.00	ref	0.00	0.00
IgG status	−0.07	0.03	0.01	−0.13	−0.01

**Table 5 animals-14-01251-t005:** Percentage with positive fecal tests for rotavirus, coronavirus, *E. coli* F5, and *C. parvum*, and mean log counts of *C. perfringens* on day 7, day 14, and day 21 in Colost-suppl. and Control calves.

Pathogen	Treatment	Day 7	Day 14	Day 21
Rotavirus	Control	0.24	0.18	0.03
Rotavirus	Colostrum	0.11	0.05	0.14
	Chi-square *p*-value	0.17	0.09	0.12
Coronavirus	Control	0.00	0.00	0.00
Coronavirus	Colostrum	0.00	0.03	0.00
	Chi-square *p*-value	n.e.	0.34	n.e.
*E. coli* F5	Control	0.00	0.00	0.00
*E. coli* F5	Colostrum	0.00	0.00	0.00
	Chi-square *p*-value	n.e.	n.e.	n.e.
*C. parvum*	Control	0.24	0.88	0.16
*C. parvum*	Colostrum	0.28	0.68	0.17
	Chi-square *p*-value	0.68	0.04	0.91
*C. perfringens*	Control	1.38	2.32	1.44
*C. perfringens*	Colostrum	1.69	1.78	1.53
	JT *p*-value	0.42	0.23	0.76

**Table 6 animals-14-01251-t006:** Total colostral IgG and serum IgG (in mg/mL) on days 2, 7, 14, and 21 and percentage inhibition in indirect ELISA for coronavirus, rotavirus, and *E. coli* F5 antibodies in first colostrum feed and in serum on days 2, 7, 14, and 21.

IgG	Group	Colostrum	Day 2	Day 7	Day 14	Day 21
Total	Control	89.2	21.3	19.8	16.2	14.7
Total	Colost-suppl.	89.2	21.4	20.1	17.3	15.5
Total	*p*-value	0.86	0.43	0.24	0.64	0.71
Corona	Control	77.7	93.3	92.6	90.6	87.4
Corona	Colost-suppl.	73.8	91.0	89.5	85.9	84.1
Corona	*p*-value	0.86	0.95	0.79	0.38	0.44
Rotavirus	Control	87.0	81.7	76.5	72.7	66.0
Rotavirus	Colost-suppl.	78.7	82.5	77.8	71.9	67.9
Rotavirus	*p*-value	0.29	0.68	0.89	0.91	0.97
*E. coli* F5	Control	89.0	86.6	90.0	88.9	85.0
*E. coli* F5	Colost-suppl.	80.9	84.1	83.4	85.4	86.3
*E. coli* F5	*p*-value	0.24	0.43	0.24	0.64	0.71

## Data Availability

The datasets presented in this article are not readily available because the data are part of an ongoing study.

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
