# Peer review of "The Effect of Colostrum Supplementation during the First 5 Days of Life on Calf Health, Enteric Pathogen Shedding, and Immunological Response"

_animals, 2024, doi:10.3390/ani14081251_

Round 1

Reviewer 1 Report

Comments and Suggestions for Authors

Dear authors, 

Please find below my notes about some aspects of your manuscript:

Abstract

Row 56-58; 59-60 [….] A non-significant trend….It is advisable to avoid such expressions as they may mislead readers; an observation is either significant or not.

Introduction

The introduction is thoroughly documented; however, I wish to would like to highlight within the text the novelty of your study compared to other similar studies.

Rows 78- 85: Please mention references

Material and Methods:

Mention the protocol experiment (no, issued, approved by…..)

Provide information about cage dimensions

Row 135: ad libitum (italic, please)

Rows 148, 158 – farm protocols? Please mention at the beginning in Material and Methods chapter the farm protocol no, issued by, approved by….., as requested previously.

Row 207: Versus.Use italics.

Results: Row 216-218 I would recommend to place the phrase in Material and Methods chapter

For subtitle 3.2 and 3.3 and also check the entire manuscript please verify the guideline  format for Animals submission…..it is either with small or capital letters!

Row 237: Font for the footnote of the table? Check it, please.

Table 3. You mention : Resp first week/ preweaning (short name after respiration, I presume). Also, when referring to OR/Lower CC/Higher CI in table titles, it is essential to include all short names or abbreviations in the tables' footnotes, even if they have been previously mentioned in the text or other tables. Check the font inside table 3, please!!

Rows 402-403: [….] increased the concentrations of serum immunoglobulins, triacylglycerols, cholesterol and total protein. You have no data about the parameters you mentioned into the manuscript Please add values and comments regarding triacylglycerols, cholesterol and total protein maybe glucose.

Thank you

Author Response

Dear Reviewer, Thank you for the time you took to review our paper. I hope that I have responded to all of your concerns. 

Abstract
Row 56-58; 59-60 [….] A non-significant trend….It is advisable to avoid such expressions as they may mislead readers; an observation is either significant or not.
Answer: Removed ‘non-significant’

Introduction
The introduction is thoroughly documented; however, I wish to would like to highlight within the text the novelty of your study compared to other similar studies.
Answer: We have added some sentences, about the additional value of this study in lines 91-94. 
Rows 78- 85: Please mention references
 Answer: References included. 
Material and Methods:
Mention the protocol experiment (no, issued, approved by…..)
This study was initiated by Dr Anna Catharina Berge. It was approved by MSD-AH through number MSS-4147. The editors have received additional information regarding the study approval. 

Provide information about cage dimensions
Answer: The dimensions of the Plexiglas hutches and group pens have been included in M&M
Row 135: ad libitum (italic, please)
Answer: Corrected
Rows 148, 158 – farm protocols? Please mention at the beginning in Material and Methods chapter the farm protocol no, issued by, approved by….., as requested previously.
Answer : This information has been provided to the editor. If the editor indicates that this information needs to be publicly available, then we will modify the text to include this information. 
Row 207: Versus.Use italics.
Answer: Corrected

Results: Row 216-218 I would recommend to place the phrase in Material and Methods chapter
Answer: Sentence deleted. The calf numbers are stated in M&M and now also in shortened abstract. 
For subtitle 3.2 and 3.3 and also check the entire manuscript please verify the guideline  format for Animals submission…..it is either with small or capital letters!
Answer: Verified. 
Row 237: Font for the footnote of the table? Check it, please.
Answer: We will work with editors on this one. SAS program does not have Palatino Linotype font as an option. 
Table 3. You mention : Resp first week/ preweaning (short name after respiration, I presume). Also, when referring to OR/Lower CC/Higher CI in table titles, it is essential to include all short names or abbreviations in the tables' footnotes, even if they have been previously mentioned in the text or other tables. Check the font inside table 3, please!!
Answer: Font in tables changed to Palatino Linotype, and footnotes placed in table
Rows 402-403: [….] increased the concentrations of serum immunoglobulins, triacylglycerols, cholesterol and total protein. You have no data about the parameters you mentioned into the manuscript Please add values and comments regarding triacylglycerols, cholesterol and total protein maybe glucose.¨
Answer: Since we did not evaluate these, I have simply removed that statement. 

Reviewer 2 Report

Comments and Suggestions for Authors

I respect the effort put into this study by the authors, who explored the analysis of colostrum on calf health and weight gain through multiple perspectives, which helps to clarify the important role of colostrum in calf health protection. However, some flaws still need to be clarified:

1) The abstract is too wordy, explaining too many treatments (which should be in the methods section) and not focused enough.

2) The Methods section is too confusing to figure out how to handle the experimental and control groups. Instead, there is too much words about experimental treatment in the abstract and it is recommended to put it in the methods.

3) If colostrum is beneficial to gut health and pathogen resistance, then the background should appropriately show previous reports on how colostrum protects intestinal health, but this manuscript does not state it in the background.

4) The authors examined the differences in body weight, fecal pathogens, etc. between the treatment and control groups and did not detect any differences between two groups, but the authors did not explain this lack of difference between the two groups in the Results and Discussion section in a sufficiently reasonable manner (why there is no difference, is it because of similar nutrients in the treatment group and the control group, or something else), which is not appropriate. For example, ‘no significant increase in weight gain was seen in the colostrum-supplemented calves (L333)’, the authors speculated that ‘the lack of effect of weight gain in our colostrum-supplemented was most likely to the overall high level of gut health in the calves and possibly low pathogen exposure levels’. If so, there is no necessary to conduct the following experiment (such as gut health and fecal enteric pathogens) because calves were in good health in both groups. 

5) In the case of this study, which result would you prefer, a result with significant difference in the treatment group (the colostrum supplementation treatment) than in the control group (milk-derived supplement with similar protein and fat composition as the colostrum supplement), thus emphasizing the protective effect of colostrum on calf health, or would you prefer not to have a difference in the results between the two groups, thus emphasizing that the control group has the same protective effect on calf health, maintains similar body weights, and has the potential to replace the treatment group (colostrum)? Which result would you prefer? I would like to express that either result is based on the results of this study, but the manuscript should have a clear objective and a reasonable explanation of the results needs to be given (why there is or there is no difference between two groups).

Others:

L82. ) is omitted here..

L120. ‘24,5 months’, is this 24.5 or 24-25 months?

L181. Significant. Significance.

L242. Add a ‘.’ after (P <0.01). Same error in L398.

Fig2. I would suggest add * as different significance in the histogram.

Author Response

I respect the effort put into this study by the authors, who explored the analysis of colostrum on calf health and weight gain through multiple perspectives, which helps to clarify the important role of colostrum in calf health protection. However, some flaws still need to be clarified:
1) The abstract is too wordy, explaining too many treatments (which should be in the methods section) and not focused enough.
Answer: The abstract has been shortened to under 200 words according to Animals’ format. 
2) The Methods section is too confusing to figure out how to handle the experimental and control groups. Instead, there is too much words about experimental treatment in the abstract and it is recommended to put it in the methods.
Answer: We have modified M&M to make the study protocol clearer. 
3) If colostrum is beneficial to gut health and pathogen resistance, then the background should appropriately show previous reports on how colostrum protects intestinal health, but this manuscript does not state it in the background.
Answer: We have references to these studies in Line 89, with references to Roffler et al, 2003, Gauthier et al, 2006 and Ontsouka et al, 2016. We further include Blättler et al, 2001. 
4) The authors examined the differences in body weight, fecal pathogens, etc. between the treatment and control groups and did not detect any differences between two groups, but the authors did not explain this lack of difference between the two groups in the Results and Discussion section in a sufficiently reasonable manner (why there is no difference, is it because of similar nutrients in the treatment group and the control group, or something else), which is not appropriate. For example, ‘no significant increase in weight gain was seen in the colostrum-supplemented calves (L333)’, the authors speculated that ‘the lack of effect of weight gain in our colostrum-supplemented was most likely to the overall high level of gut health in the calves and possibly low pathogen exposure levels’. If so, there is no necessary to conduct the following experiment (such as gut health and fecal enteric pathogens) because calves were in good health in both groups. 
Answer: You are correct, we were also disappointed that there was not more enteric disease detected during the study. This was not expected, as the farm had a history of cryptosporidiosis in the calves. However, we feel that if studies are only published that show significant improvement, then there is publication bias. It is important to also publish studies where there are limited improvements in the digestive health of the calves. Therefore, we feel that this study contributes to the knowledge regarding post-closure colostrum supplementation. 
5) In the case of this study, which result would you prefer, a result with significant difference in the treatment group (the colostrum supplementation treatment) than in the control group (milk-derived supplement with similar protein and fat composition as the colostrum supplement), thus emphasizing the protective effect of colostrum on calf health, or would you prefer not to have a difference in the results between the two groups, thus emphasizing that the control group has the same protective effect on calf health, maintains similar body weights, and has the potential to replace the treatment group (colostrum)? Which result would you prefer? I would like to express that either result is based on the results of this study, but the manuscript should have a clear objective and a reasonable explanation of the results needs to be given (why there is or there is no difference between two groups).
The null hypothesis is that there is no difference between Colostrum-supplementation and the Control-supplementation. The alternative hypothesis that we are attempting to show is that there is a significant difference between colostrum supplementation and no colostrum supplementation post-closure of the gut. The control supplementation is to assure that this difference is not due to the additional energy and nutrients supplied through the colostrum, but due to the biological active components in the colostrum. We add a statement regarding the null hypothesis in the material and methods statistical analysis section. 
Others:
L82. ) is omitted here..Corrected 
L120. ‘24,5 months’, is this 24.5 or 24-25 months? 24.5
L181. Significant. Significance. Corrected
L242. Add a ‘.’ after (P <0.01). Same error in L398. Corrected

Fig2. I would suggest add * as different significance in the histogram. *Inserted
;;3
In the MM, it is not specified that calves were chosen over a period of time and not at once. we only see this in the result section. therefore, what experimental design was used for this experiment? time effect needs to be considered in the statistical analysis. Also, its is not clear whether the calves came from the same farm or from many different farms. the reader guess that calves came from only one farm since the health status was good overall. THe authors need to precise this point in their MM.
Answer: In the abstract and the first paragraph of the M&M we state that this is ‘a farm’. We make it clearer by changing text from ‘a farm’ to ‘one’ farm and include more statements talking about ‘this farm study’. The calves were enrolled over a long period from Nov 20 2019 to March 23 2023, we have mentioned these dates in text. The sample size is too small to evaluate significant seasonal and temporal trends, but as the calves were randomly enrolled, there were calves allocated in all years and seasons and approx. equal proportions. Furthermore, we evaluated seasonal effect in multivariate analysis and they were not significantly associated with any study outcome.  
Section 4.4. There was a significant effect of colostrum fed calves. The authors dis not discuss this important result in relation to other studies. this section would need to be further discussed in more depth.
Answer: We include a reference to the study by Kargar et al. who also noted less days of fever and less days of depressed attitude in calves supplemented with colostrum for 14 days. 
L102 change "...disease days with 40%..." for " ....disease days by 40%...." Done
Answer: Corrected

Reviewer 3 Report

Comments and Suggestions for Authors

In the MM, it is not specified that calves were chosen over a period of time and not at once. we only see this in the result section. therefore, what experimental design was used for this experiment? time effect needs to be considered in the statistical analysis. Also, its is not clear whether the calves came from the same farm or from many different farms. the reader guess that calves came from only one farm since the health status was good overall. THe authors need to precise this point in their MM.

Section 4.4. There was a significant effect of colostrum fed calves. THe authors dis not discuss this important result in relation to other studies. this section would need to be further discussed in more depth.

L102 change "...disease days with 40%..." for " ....disease days by 40%...."

Comments on the Quality of English Language

Laguage quality is sufficient and no major concern was noted

Author Response

Answer: Dear Reviewer, Thank you for your valuable review. We have rewritten text to ensure that your questions are not of concern for future readers. 

In the MM, it is not specified that calves were chosen over a period of time and not at once. we only see this in the result section. therefore, what experimental design was used for this experiment? time effect needs to be considered in the statistical analysis. Also, its is not clear whether the calves came from the same farm or from many different farms. the reader guess that calves came from only one farm since the health status was good overall. THe authors need to precise this point in their MM.
Answer: In the abstract and the first paragraph of the M&M we state that this is ‘a farm’. We make it clearer by changing text from ‘a farm’ to ‘one’ farm and include more statements talking about ‘this farm study’. The calves were enrolled over a long period from Nov 20 2019 to March 23 2023, we have mentioned these dates in text. The sample size is too small to evaluate significant seasonal and temporal trends, but as the calves were randomly enrolled, there were calves allocated in all years and seasons and approx. equal proportions. Furthermore, we evaluated seasonal effect in multivariate analysis and they were not significantly associated with any study outcome.  
Section 4.4. There was a significant effect of colostrum fed calves. The authors dis not discuss this important result in relation to other studies. this section would need to be further discussed in more depth.
Answer: We include a reference to the study by Kargar et al. who also noted less days of fever and less days of depressed attitude in calves supplemented with colostrum for 14 days. 
L102 change "...disease days with 40%..." for " ....disease days by 40%...." Answer: Corrected

Reviewer 4 Report

Comments and Suggestions for Authors

GENERAL COMMENTS

The manuscript by Berge and collaborators reports the effects of colostrum supplementation from mothers vaccinated against rotavirus, coronavirus and Escherichia coli F5 and F41, on the health, immunity and performance of pre-weaned Holstein calves during five days after birth.

The topic, although it has already been extensively investigated by researchers, I believe it can still bring benefit to the topic. The experimental plan seems good to me, as does the statistical analysis.

My doubts concern more than anything else the data collection which took place in experimental moments which were very distant from each other and which may therefore include other variables.

Information such as the lactation schedule and the types of starter and hay available to the calves should be added to the materials and methods.

In the discussions I would explain better the poor effect of colostrum supplementation on immunity, while in the conclusions I would be more cautious (in light of the results obtained).

Throughout the text the characters used must be standardized and spaces must be controlled.

SPECIFIC COMMENTS

L16: Insert space after the colon

L30-64 Please summarize the abstract a little. According to the magazine's standards, it must be a maximum of 200 words. Redefine the objective better, example: ....compared to unfed calves... specify better…not fed WITH ?? Furthermore, it is not clear what type of colostrum the control group takes.

L 80: eliminate extra space

L 81-84: L24: add the reference

L 90: eliminate extra space

L119-120: ......The average age of first calving was 24,5 months..... also enter the Standard Deviation

Please specify the calving order of the mothers of the calves enrolled in the study. Were they all primiparous? or multiparous? second or third calving?

How much starter, concentrate and hay were fed to the calves?

L132: Describe the lactation curve used in weaning the observed calves (not just the maximum amount of milk).

L 216-2017: The calves were observed over several years (November 2019-March 2023) and not all at the same time in the same period. So how did you exclude the variables related to weather, season, company management etc??

Uniforms the characters in table 3.

Check that the results in paragraph 3.3 and Table 5 and Figure 2 correspond to each other (names of bacteria etc.).

Insert the asterisk in figure 2 where there is significance.

Insert the asterisk in figure  3  where there is significance.

Check table 6. It's not clear

L344-346 Could it be due to the greater feed consumption in one group compared to the other?

L 412 The temperature in the control calves was 38.8 while in the Colost_supp.calves calves it was 38.9 at the same dv. Although significant, I advise the authors to be more cautious and focus less on this minimal difference which could therefore have little relevance at a clinical and diagnostic level.

Author Response

Answer: Dear reviewer. I appreciate your really detailed review of our document. I can see that you have put time and effort and thought into it...and this is what peer-review is all about. Thank you for that. I hope that I have addressed your concerns and points, so that future readers will not sit with similar questions.

For your minor text comments. I have just put a 'Done' to indicate that your comment was noted and corrected. 

My doubts concern more than anything else the data collection which took place in experimental moments which were very distant from each other and which may therefore include other variables.
Answer: We are indeed aware of the extended time-period for this study. This is the disadvantage of conducting studies on European farms that generally are more family run farms. The Veterinary clinic that conducted all evaluations ensured consistent evaluations of the calves. Since both treatment groups were enrolled over the whole study period, there is likely minimal influence of environmental factors. This is of course always a possibility. We include a sentence about this in the discussion. 
Information such as the lactation schedule and the types of starter and hay available to the calves should be added to the materials and methods.
Answer: We have elaborated on the lactation schedule in M&M. The first starter offered was ad lib a muesli starter up to 4 weeks of age, and thereafter ad lib an all-pelleted grain starter supplemented with hay until weaning. After one month of age, the calves were placed in group pens of 3 calves, and there the treatment groups were mixed in the group pens. 
In the discussions I would explain better the poor effect of colostrum supplementation on immunity, while in the conclusions I would be more cautious (in light of the results obtained).
Answer: Our study confirms that there is no enhanced systemic antibody immunity with post-closure supplementation of colostrum. However, our study indicates that the local gut immunity to enteric pathogens such as Cryptosporidia and rotavirus is enhanced with post-closure colostrum supplementation, and this has also been indicated in other studies. We have toned down the conclusions. 
Throughout the text the characters used must be standardized and spaces must be controlled.
Answer: We have ensured that all tables are now in Palatino Lintoype size 10. 
SPECIFIC COMMENTS
L16: Insert space after the colon. Done
L30-64 Please summarize the abstract a little. According to the magazine's standards, it must be a maximum of 200 words. Redefine the objective better, example: ....compared to unfed calves... specify better…not fed WITH ?? Furthermore, it is not clear what type of colostrum the control group takes.
Answer: We have shortened the abstract to under 200 words. The control calves receive a milk replacer derived with similar protein and fat composition as colostrum, but devoid of the colostral immunoglobulins and other colostral components that are not found in same levels in milk. This control supplement was colored (to ressemble colostrum) and frozen similarly to the colostrum so that the study personnel were blinded and there was no difference in nutritional levels of the two treatment groups. 
L 80: eliminate extra space. Done
L 81-84: L24: add the reference Done
L 90: eliminate extra space Done
L119-120: ......The average age of first calving was 24,5 months..... also enter the Standard Deviation
I do not believe this herd data necessitates standard deviations, as this is just describing the general herd performance. 
Please specify the calving order of the mothers of the calves enrolled in the study. Were they all primiparous? or multiparous? second or third calving?
Answer: We indicated that they were enrolled from all parity cows. Calves were randomly assigned to treatment using a random number list, and parity was evaluated as a covariate in multivariable models, and was found to be not significantly associated with any outcomes analyzed. It should be noted that with the limited study size, the ability to detect subtle impact of factors such as parity and season may have been limited. 
How much starter, concentrate and hay were fed to the calves? Calves were fed ad libitum starter and hay. This was not measured in the study. 
L132: Describe the lactation curve used in weaning the observed calves (not just the maximum amount of milk). We have elaborated on milk feeding strategy in M&M. 
L 216-2017: The calves were observed over several years (November 2019-March 2023) and not all at the same time in the same period. So how did you exclude the variables related to weather, season, company management etc??
Answer:The calves were randomly enrolled during these various season and year, thus weather, season and company management occurred equally in the two groups. I evaluated the effect of season as a covariate in the multivariable analysis, but it was found to be non-significant. 
Uniforms the characters in table 3. Done
Check that the results in paragraph 3.3 and Table 5 and Figure 2 correspond to each other (names of bacteria etc.). Done
Insert the asterisk in figure 2 where there is significance. Done
Insert the asterisk in figure  3  where there is significance. Done
Check table 6. It's not clear. The title has been revised to make table clearer. 
L344-346 Could it be due to the greater feed consumption in one group compared to the other? 
Answer: This could also be a factor. In general gut health impacts starter grain intake negatively, but since we did not have a significant impact on diarrhoea scores by using colostrum supplementation (which has been seen in other studies) it is likely that starter grain intake was not significantly impacted by the colostrum supplementation (as was noted by Berge et al. 2009). 

L 412 The temperature in the control calves was 38.8 while in the Colost_supp.calves calves it was 38.9 at the same dv. Although significant, I advise the authors to be more cautious and focus less on this minimal difference which could therefore have little relevance at a clinical and diagnostic level.
This minimal difference as measured and evaluated using linear models and comparisons of mean temperatures is due to that there were more days with elevated temperatures (light fever) in calves in the control calves than the colostrum-supplemented calves. I chose to work with recorded temperatures, rather than to make categorical variables of body temperature. The days of elevated temperatures in the calves corresponded to days when the calves were noted to have depressed attitude or respiratory signs.  

Round 2

Reviewer 2 Report

Comments and Suggestions for Authors

The authors have addressed all comments. 

Author Response

I would like to thank reviewer 2 for taking a second look at our document. We have scrutinized the document again to assure that the formatting and text is as good as possible. 

Reviewer 4 Report

Comments and Suggestions for Authors

The authors responded to my comments, however I recommend reviewing the entire manuscript and in particular the structure of the tables and captions.

Author Response

I would like to thank reviewer 4 for taking a second look at our document. We have scrutinized the document again to assure that the formatting and text is as good as possible. Indeed I found that he histogram of serum-specific antibodies was not correct, it was missing two sampling day for E. coli, and I have replaced that.